# Brainstem network dynamics underlying the encoding of bladder information

Anitha Manohar, Andre L Curtis, Stephen A Zderic, Rita J Valentino*

The Children's Hospital of Philadelphia, University of Pennsylvania, Philadelphia, United States

**Abstract** Urodynamic status must interact with arousal and attentional processes so that voiding occurs under appropriate conditions. To elucidate the central encoding of this visceral demand, multisite recordings were made within a putative pontine-cortical micturition circuit from the pontine micturition center (PMC), locus coeruleus (LC) and medial prefrontal cortex (mPFC) during cystometry in unanesthetized rats. PMC neurons had homogeneous firing patterns, characterized by tonic activity and phasic bursts that were temporally associated with distinct phases of the micturition cycle. LC and cortical activation became synchronized 20-30 s prior to micturition. During this pre-micturition interval, a theta oscillation developed in the LC, the mPFC desynchronized and LC-mPFC coherence increased in the theta frequency range. The temporal offset between the shift in LC-mPFC network activity and micturition may allow time to disengage from ongoing behaviors unrelated to micturition and initiate specific voiding behaviors so that micturition occurs in environmentally and socially appropriate conditions.

DOI: https://doi.org/10.7554/eLife.29917.001

## Introduction

The physiological elimination of urine, micturition, must occur in socially appropriate and environmentally safe conditions. In some species micturition is integral to social communication, for example, with regard to sexual status and social rank (*Ralls, 1971*; *Desjardins et al., 1973*). These neurobehavioral aspects of micturition demand precise temporal coordination between bladder activity and the initiation of motor patterns that characterize voiding behavior. For example, bladder filling should increase arousal and shift the focus of attention before pressure reaches the micturition threshold so that ongoing behavior is suspended and voiding behavior is initiated prior to the passage of urine. A lack of coordination between voiding behaviors and micturition characterizes dysfunctional voiding syndromes, such as enuresis that can occur both in children and older adults (*Torrens and Collins, 1975*; *Nevéus, 2017*). This temporal coordination is performed by circuits that convey reciprocal communication between the brain and bladder.

The pontine micturition center (PMC) is central to a circuit that is positioned to coordinate neurobehavioral and visceral limbs of micturition (*Valentino et al., 2011b*). Through its projections to lumbosacral preganglionic neurons that regulate the parasympathetic innervation of the detrusor, the PMC is proposed to function as an all or none switch, whereby the neurons become activated at some threshold pressure and engage the descending limb that initiates bladder contraction (*de Groat, 2006*). In contrast to our knowledge of how the brain regulates bladder function, there is a paucity of information on the bottom-up processing of bladder information that is necessary for the neurobehavioral limb of the micturition reflex. Human brain imaging studies using SPECT, PET and fMRI have guided our current thinking about the structures involved in this complex process (*Griffiths, 2004*; *Griffiths et al., 2009*). However, these approaches have limitations with respect to temporal and spatial resolution. Using anatomical and physiological approaches, we described one circuit that is positioned to convey bladder information to cortical regions involved in executive

*For correspondence:
valentinorj@nida.nih.gov

Competing interests: The authors declare that no competing interests exist.

**eLife digest** How do we know when we need to find a bathroom? As the bladder fills up, it sends signals to the brain to say that it needs emptying. But before the brain sends a message back to the bladder muscles telling them to contract to release urine, it first triggers a change in behavior. By increasing our alertness and arousing our senses, the brain ensures that we begin to look for a place where it is safe and appropriate to urinate. Only when we have found such a place will the brain tell the bladder to empty.

Previous work has suggested that two brain regions play important roles in this process: the pontine micturition center (PMC) and its neighbor, the locus coeruleus. The PMC is thought to act as an on-off switch. When the bladder reaches a certain level of fullness the PMC activates, which tells the bladder muscles to contract. The locus coeruleus helps animals pay attention to important stimuli by making them more alert and energized whenever such stimuli are present.

By recording the activity of neurons in the brains of rats while also measuring the pressure inside their bladders, Manohar et al. show that the PMC and the locus coeruleus work together to coordinate behavior and bladder emptying. Filling the bladder causes neurons in the locus coeruleus to activate in synchronized waves. This helps the locus coeruleus communicate with the brain's outer layer, the cortex, leading to an increase in sensory alertness and arousal. This all happens before the bladder reaches the threshold fullness that activates the PMC, explaining why behavioral changes occur before urination. Manohar et al. show too that PMC neurons also activate when the rat is not urinating, suggesting that the PMC is more than an on-off switch.

Healthy people experience the sensation of needing to empty their bladder well before the bladder is full, but people who do not receive these sensory signals may be unable to tell when they need to take action. This can lead to bedwetting in children and to incontinence in the elderly. Targeting the brain circuit that responds to bladder signals could lead to new treatments for these conditions.

DOI: https://doi.org/10.7554/eLife.29917.002

function. We identified a population of spinal-projecting PMC neurons that are retrogradely labeled from both the lumbosacral spinal cord and the pontine noradrenergic nucleus locus coeruleus (LC) (*Valentino et al., 1996*). The LC has a widely distributed axonal network that extensively innervates the forebrain and particularly the cortex where norepinephrine serves as a neuromodulator of cortical activity (*Swanson and Hartman, 1975*). Selective pharmacological or optogenetic activation of LC neurons or activation using Gs-linked DREADDS is sufficient to produce cortical electroencephalographic (EEG) activation indicative of arousal (*Berridge and Foote, 1991*; *Carter et al., 2010*; *Vazey and Aston-Jones, 2014*). Notably, LC neurons are activated by bladder and colonic distention and this is temporally linked to cortical EEG activation (*Elam et al., 1986*; *Page et al., 1992*; *Lechner et al., 1997*). In addition to increasing arousal in response to salient stimuli, LC activation is thought to reset attentional networks to shift the focus of attention to salient stimuli (*Aston-Jones and Bloom, 1981a*; *Aston-Jones and Cohen, 2005*; *Bouret and Sara, 2005*). Integrating these findings, we proposed a model whereby increases in bladder pressure could activate PMC neurons that project to the LC and the impact on LC-cortical projections would increase arousal, reorient attention and promote the initiation of a voiding behavioral repertoire while suspending incompatible ongoing behaviors (*Valentino et al., 1999*; *Valentino et al., 2011a*). Thus, through collateral projections to the LC and spinal cord based on dual retrograde labeling, the PMC could coordinate visceral and neurobehavioral limbs of the micturition reflex.

To begin to test components of this model we recorded unit activity from PMC neurons and/or LC neurons along with local field potentials (LFP) from the medial prefrontal cortex (mPFC) simultaneously with urodynamic endpoints measured during in vivo cystometry in unanesthetized rats (*Figure 1A,B*, *Figure 1—figure supplement 1–1*).

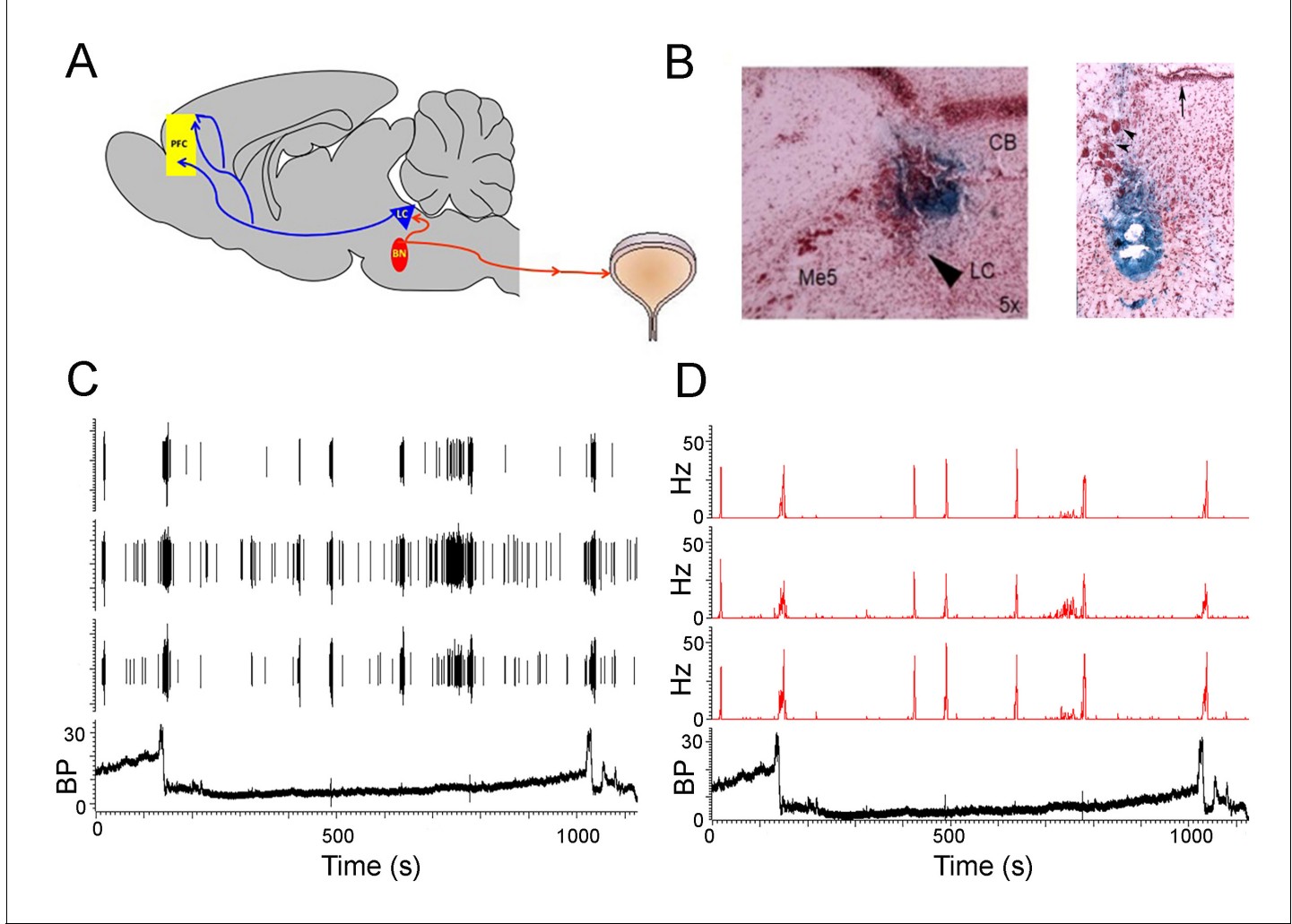

**Figure 1.** PMC neuronal activity during micturition cycles in unanesthetized rats. (**A**) Schematic showing proposed circuitry engaged during micturition and recording sites. The PMC (red oval) projects to lumbosacral preganglionic parasympathetic neurons that give rise to cholinergic input to the detrusor and produce contraction. A population of PMC spinal-projecting neurons are retrogradely labeled from the LC (blue triangle) which is just caudal, dorsal and lateral to the PMC. LC neurons project widely throughout the cortex. In the present study, single unit activity was recorded from PMC neurons and/or LC neurons and local field potential activity was recorded from the medial prefrontal cortex (yellow rectangle). Neuronal recordings were obtained simultaneously with in vivo cystometry recordings from an implanted bladder catheter. (**B**) Representative Prussian blue histological verification of a recording in the LC (left) and the PMC (right). For the LC photomicrograph CB (Cerebellum), Me5 (mesencephalic trigeminal nucleus) and the arrow points to the LC. For the PMC photomicrograph, the arrow points to the ventricle and the arrowheads point to the mesencephalic trigeminal nucleus. *Figure 1—figure supplement 1* shows a section slightly rostral to this with a lesion created by the wires (**C**) Raw waveform traces showing spikes from three PMC neurons and simultaneously recorded bladder pressure (BP mm Hg) during two micturition cycles. (**D**) Ratemeter records of the same cells shown in C.

DOI: https://doi.org/10.7554/eLife.29917.003

The following figure supplements are available for figure 1:

**Figure supplement 1.** Section through the pons that is just rostral to that shown in *Figure 1B* (PMC).
DOI: https://doi.org/10.7554/eLife.29917.004
**Figure supplement 2.** The same bladder pressure trace as shown in *Figure 1C* is aligned with a trace noting the timing of micturition (bottom panel).
DOI: https://doi.org/10.7554/eLife.29917.005

## Results

### Relationship between PMC neuronal activity and bladder pressure

After the first micturition cycle, subsequent cycles in rats became regular, with bladder pressure gradually increasing to a peak, at which point micturition occurred and bladder pressure fell to 0 mm Hg (*Figure 1—figure supplement 1–2*). Single unit activity was recorded from 36 PMC neurons from three rats during in vivo cystometry. *Figure 1C,D* shows activity from three single units recorded on one channel in the PMC during two micturition cycles. Whether depicted as the raw waveform trace (*Figure 1C*) or rate (*Figure 1D*), the pattern of PMC neuronal discharge was relatively homogeneous and this was true across all three rats (*Figures 1C, D* and *2A*). The PMC has been proposed to function as a switch such that neuronal activation results in micturition (*de Groat, 2006*). Consistent with this, in all rats, PMC neurons discharged during micturition until bladder pressure decreased to zero and the bladder was empty (*Figures 1C, D* and *2A*). *Figure 2B* shows the change in discharge rate associated with micturition for the individual cells shown in the example rasters. Most neurons increased their discharge rate and no recorded neurons were inhibited at this time. All PMC neurons also exhibited burst activity at different times with respect to micturition (*Table 1*). This was particularly apparent within the 20 s period following micturition during which bursts averaging 28 ± 1 Hz frequency and 606 ± 66 ms in duration occurred (*Table 1*, *Figures 1C, 2A and C*). These post-micturition bursts were sometimes associated with a very small, transient increase in bladder pressure (*Figure 2A* asterisks). Notably, in all PMC cells similar bursts (28 ± 2 Hz, 432 ± 35 ms duration) occurred regularly during inter-micturition intervals when bladder pressure

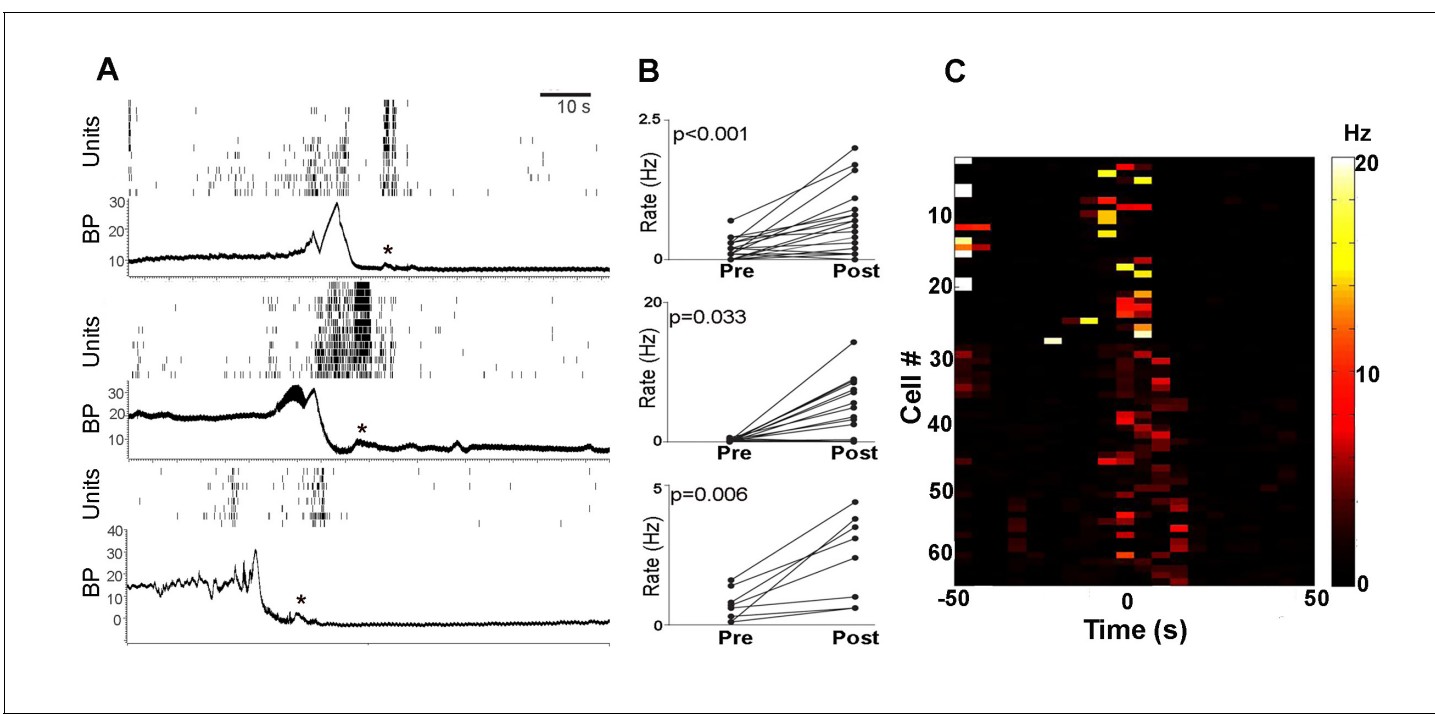

**Figure 2.** PMC neuronal activity of each rat. (A) Each panel shows PMC cell rasters and bladder pressure (BP in mm Hg) during a representative micturition cycle from each of the three rats. Note that some PMC units discharge even when BP is relatively low. Activity increases prior to peak micturition pressure and a burst of activity occurs after bladder emptying when BP returns to baseline. A slight rise in BP is temporally associated with this burst (asterisks). (B) Pairwise pre-and post-micturition firing rate changes (Hz) for individual neurons shown in the rasters in A. Pre-micturition rates were determined from the period between 40–50 s prior to peak BP and Post-micturition rates were determined from 0 to 10 s sec after peak micturition pressure. For all three panels the mean post rates were greater than mean pre-rates as indicated by the Student's t-test value (matched pairs) in the figure. (C) Heatmap of all PMC neurons recorded from the three rats showing discharge rate in Hz (coded by color scale on right) for the period of 50 s before and 50 s after micturition indicated by time = 0. For most cells, data are shown for two micturition cycles indicative of technical reproducibility. Note firing occurring 0–20 s after micturition. Bin size = 10 s.

DOI: https://doi.org/10.7554/eLife.29917.007

**Table 1.** Burst analysis of PMC neurons*

| Interval | # Bursts[†] | #spikes/burst | ISI (ms) | Duration (ms) |
|---|---|---|---|---|
| Pre-micturition | 0.3 ± 0.1 | 4.9 ± 0.2 | 54 ± 2 | 263 ± 14 |
| Intermicturition Interval | 12.3 ± 1.9 | 12.4 ± 1.2 | 38 ± 1 | 432 ± 35 |
| Post-micturition | 2.7 ± 0.5 | 18.4 ± 2.7 | 40 ± 3 | 606 ± 66 |

*values are means ± SEM determined from 36 cells from 3 rats.

[†]Mean #bursts/cell/micturition cycle

Pre-micturition = 20 s before micturition

Intermicturition interval = 120 s after first and 120 s before the following micturition

Post-micturition = 20 s after micturition

DOI: https://doi.org/10.7554/eLife.29917.006

was low (*Table 1*, *Figure 1C*). Burst activity was rare during the pre-micturition period (20 s prior to micturition). Notably, some cells were distinguished by tonic discharge between bursts (*Figure 1C*, first and third cells).

## Relationship between LC activity and bladder pressure

LC neuronal activity was recorded from 51 neurons of 4 rats. As described in previous studies in unanesthetized rats (*Aston-Jones and Bloom, 1981a*), LC neurons were spontaneously active throughout the recordings although discharge rate fluctuated with the micturition cycle. *Figure 3A, E* shows how activity of LC units of one representative rat is temporally correlated to bladder pressure. In contrast to PMC neurons, LC neurons consistently increased their discharge rate 10–30 s prior to micturition and this activation ceased with micturition (*Figure 3A,B,C,E*). The distribution of latencies between significant LC activation and micturition ranged between 0 and 65 s with most cells having a latency between 10–30 s (*Figure 3D*).

For one rat, the proximity of the LC and the PMC allowed for detection and recording of units from both regions during cystometry (*Figure 4*). Notably, the temporal relationship between LC activation, PMC neuronal activation and micturition in this particular case argued against the hypothesis that PMC neurons drive LC neurons prior to micturition because LC activation preceded PMC neuronal activation.

## LC-cortical synchrony during micturition cycles

The LC neuronal activation, occurring 10–30 s prior to micturition, was temporally correlated with mPFC desynchronization (*Figure 5A1, A2*). Simultaneous local field potential recordings (LFP) in LC and mPFC demonstrate a shift from low frequency activity at baseline (e.g., *Figure 5B1*, 0–47 s) to a prominent theta oscillation in the LC during the pre-micturition period (e.g., *Figure 5B1*, 47–77 s). This was accompanied by a desynchronization of the mPFC (*Figure 5A2, B2*). Interestingly, although the mPFC LFP magnitude was small during the 30 s period prior to micturition relative to the preceding baseline interval, there was still evidence of a theta oscillation in the mPFC during this time as well (white arrowheads in *Figure 5B2*, *Figure 6B*, cf., red and black lines). This effect was consistent as indicated by the mean power spectra of LC and mPFC LFP activity generated from each of 3 rats recorded across 8 micturition cycles (3, 2, and 3 for the individual rats) (*Figure 6A,B*). Thus, LC activity recorded from 60 to 30 s before micturition shifted from mixed low frequency activity to a prominent theta oscillation occurring 30–0 s before micturition. In concert, the mPFC power spectrum shifted from high amplitude, low frequency activity (0–5 Hz) to much lower amplitude oscillations and a decrease of power in all frequencies, characteristic of desynchronization. An increase in coherence between the LC and mPFC in the theta frequency characterized this pre-micturition interval (*Figure 6C*). This change in LC and mPFC LFP activity reverted with bladder emptying (*Figure 5*).

## Discussion

The current study combined multisite neuronal recordings with cystometry to demonstrate how neuronal activity within a pontine micturition circuit is coordinated with bladder pressure and micturition

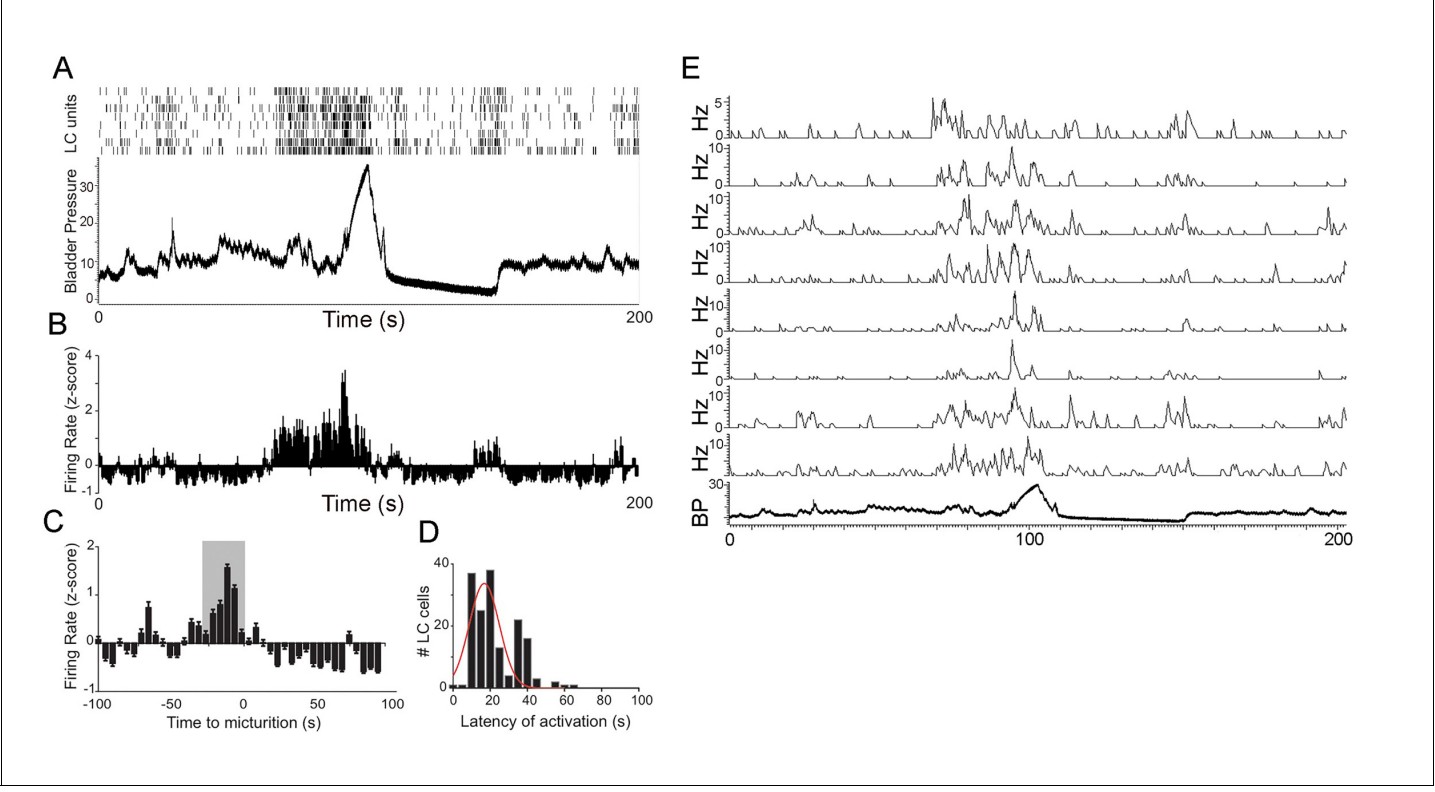

**Figure 3.** LC neuronal activity during micturition cycles in unanesthetized rats. (**A**) Spike rasters of 8 discriminated LC neurons from an individual rat and simultaneously recorded bladder pressure (mm Hg) during a micturition cycle. Abscissa indicates time in seconds. (**B**) Bars indicate the average z-score firing rate of the LC units shown in A. The abscissa in B is time locked to that in A and the firing rate is evaluated in Hz in 1 s bins. (**C**) Bar graph shows average LC firing rate from all animals (n = 4 animals, 51 neurons) across 10 micturition cycles in 5 s bins (normalized z-score). The gray box highlights the bins 30 s prior to micturition-note consistent LC activation at this time). Peak bladder pressure is at time 0 s. (**D**) Histogram showing the distribution of latencies of LC activation prior to peak micturition pressure for all neurons and across all micturition cycles. (**E**) Ratemeter records of same cells as in A.

DOI: https://doi.org/10.7554/eLife.29917.008

in the unanesthetized rat. Neuronal activity was relatively homogeneous both within the PMC and the LC. As predicted, PMC neuronal activity was temporally correlated with micturition. However, a novel finding was the consistent high frequency bursts of activity following micturition and the repeated patterns of similar bursts during the intermicturition interval when bladder pressure was low. These patterns suggest that the PMC is more than an on-off switch in the regulation of micturition. Importantly, the findings demonstrated a temporal offset between LC neuronal activation and micturition during which a theta oscillation develops in the LC, LC-mPFC coherence is increased and the mPFC becomes desynchronized. These changes in the LC-mPFC network activity may facilitate the initiation of voiding behaviors prior to micturition by increasing arousal and shifting attention towards visceral demands.

## Caveats

The present findings should be interpreted with consideration of certain technical limitations. In vivo cystometry was used to monitor and regulate micturition cycles. However, it does not mimic natural micturition cycles, which would be longer in the absence of steady bladder infusion. Although the multiwire recording technique allows for sampling activity from multiple neurons within the same nucleus, the results are inevitably biased toward activity that the wires can record from and that can be reliably discriminated. We cannot rule out the possibility that certain neuronal populations in a nucleus were not sampled. This may be a particular problem for PMC which is neurochemically heterogeneous (*Sutin and Jacobowitz, 1988*). This is less of a caveat for the LC, which is relatively

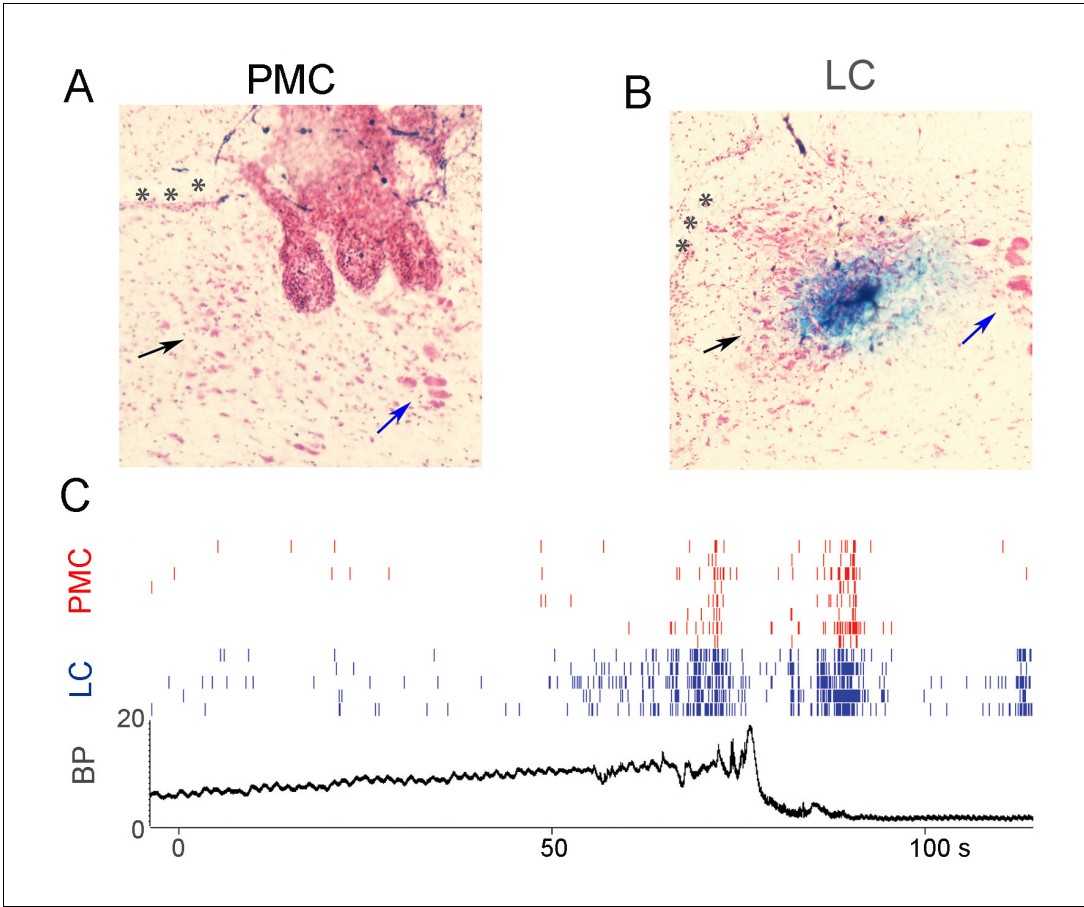

**Figure 4.** Simultaneous recordings of PMC and LC neuronal firing during micturition. (**A**) Histological verification of recording sites in the PMC. The track of the electrode wires can be seen and the most medial track impinges on the lateral part of the PMC (black arrow). Asterisks indicate the ventricle and the blue arrow points to the mesencephalic trigeminal nucleus. (**B**) Histological verification of the recording site in the LC of the same rat which is apparent as the Prussian blue reaction product. Arrows and asterisks as in A. (**C**) Spike rasters of multiple BN neurons (red) and LC neurons (blue) recorded from the same rat during a micturition cycle. The lower panel shows bladder pressure recorded simultaneously. Note that LC neuronal activation precedes PMC neuronal activation and both occur prior to peak micturition pressure. Likewise, after bladder pressure returns to baseline, LC neurons become activated and this precedes activation of PMC neurons.

DOI: https://doi.org/10.7554/eLife.29917.009

homogeneous in the rat (*Aston-Jones et al., 1995*). Only females were used in the study because the bladder surgery is somewhat easier. The surgery was relatively complex because of the need to electrophysiologically localize brain electrodes as well as to implant the catheter, so that we chose to use females. Therefore, there is a caveat that our interpretations may not extend to both sexes. However, we previously demonstrated the same temporal correlation between cortical activation and peak bladder pressure in male rats (*Kiddoo et al., 2006*; *Rickenbacher et al., 2008*).

## PMC neuronal activity

Most previous studies of PMC neuronal activity with relation to bladder pressure were done in anesthetized rats or cats with the bladder distended under isotonic conditions such that reflex contractions would occur (*Willette et al., 1988*; *Tanaka et al., 2003*; *de Groat and Wickens, 2013*). Three general neuronal types were identified in these conditions, neurons that discharge before and during reflex bladder contractions, neurons that are inhibited during contractions and active between contractions and neurons that fire transiently at the beginning of the contractions. The majority of recorded neurons in those studies were of the second type. One study recorded activity throughout

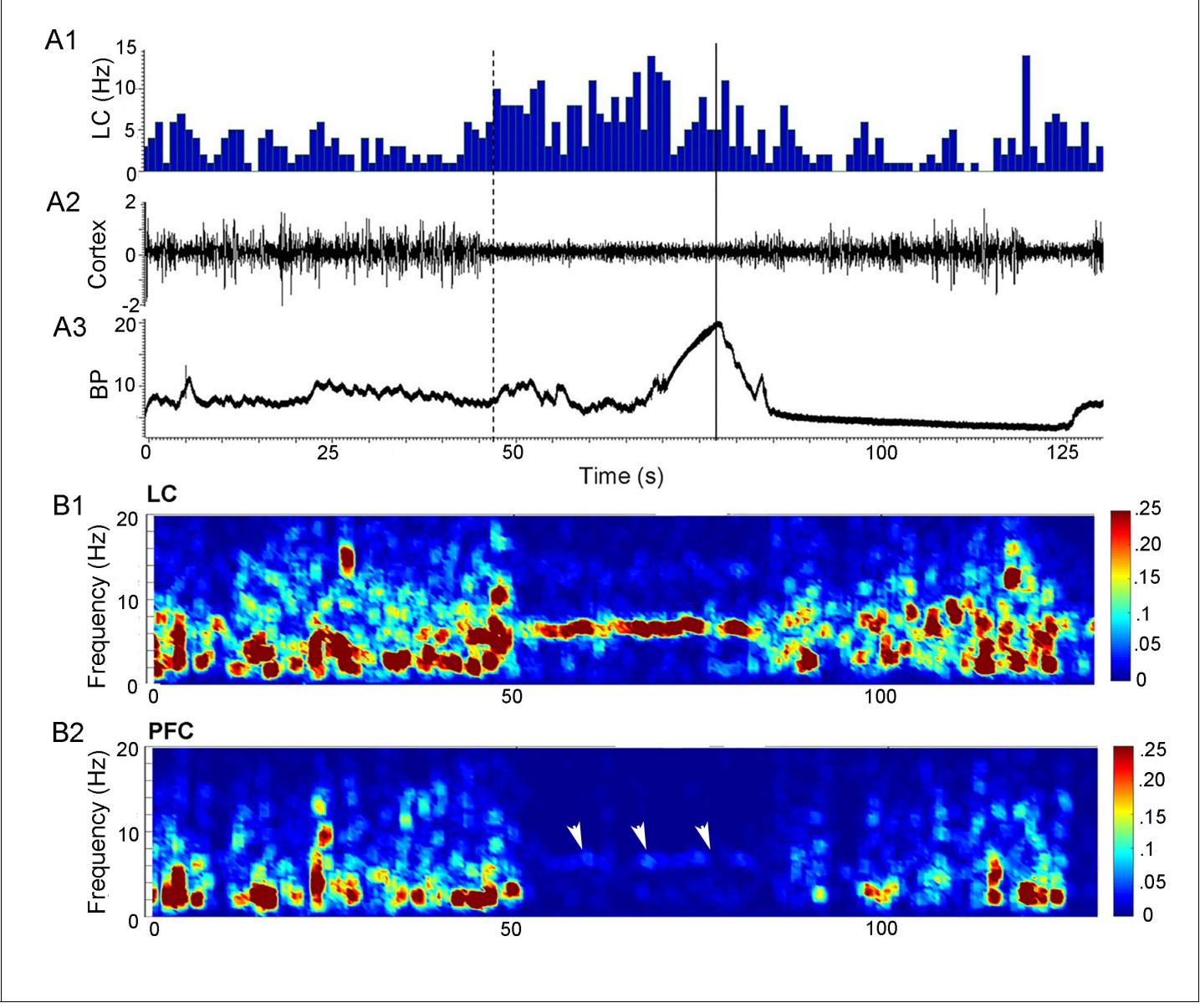

**Figure 5.** Synchronization between LC and cortical network activity prior to micturition. (**A**) A1 shows a ratemeter record of a single LC unit (bin size = 1 s). A2 is the raw cortical LFP trace. A3 shows bladder pressure (BP, mm Hg) with the solid line indicating peak micturition pressure. The dotted line indicates 30 s before peak micturition pressure. Note that LC activation is temporally correlated with cortical desynchronization and both occur just over 30 s prior to peak micturition pressure. (**B**) Heat maps showing power in different frequency bands (0–20 Hz, y-axis) with respect to time in LC (**B1**) and mPFC (**B2**) from same rat as shown in A. The abscissae indicate time (s) and are aligned to correspond to abscissae in A. The LC LFP shows a prominent theta rhythm prior to peak micturition pressure that corresponds to cortical desynchronization. White arrowheads in B2 point to a dim band in the theta frequency range of the mPFC LFP.

DOI: https://doi.org/10.7554/eLife.29917.010

the pons and medulla during cystometry in the absence of anesthesia in the decerebrate cat (*Sugaya et al., 2003*). This study also reported neurons that were either activated, inhibited or unaffected during the contraction with the majority being inhibited. Notably, in that study the mapping of these neuronal types demonstrated that nearly all neurons that were inhibited during bladder contraction were ventral to the region corresponding to the PMC, whereas most neurons within the PMC were activated during bladder contraction.

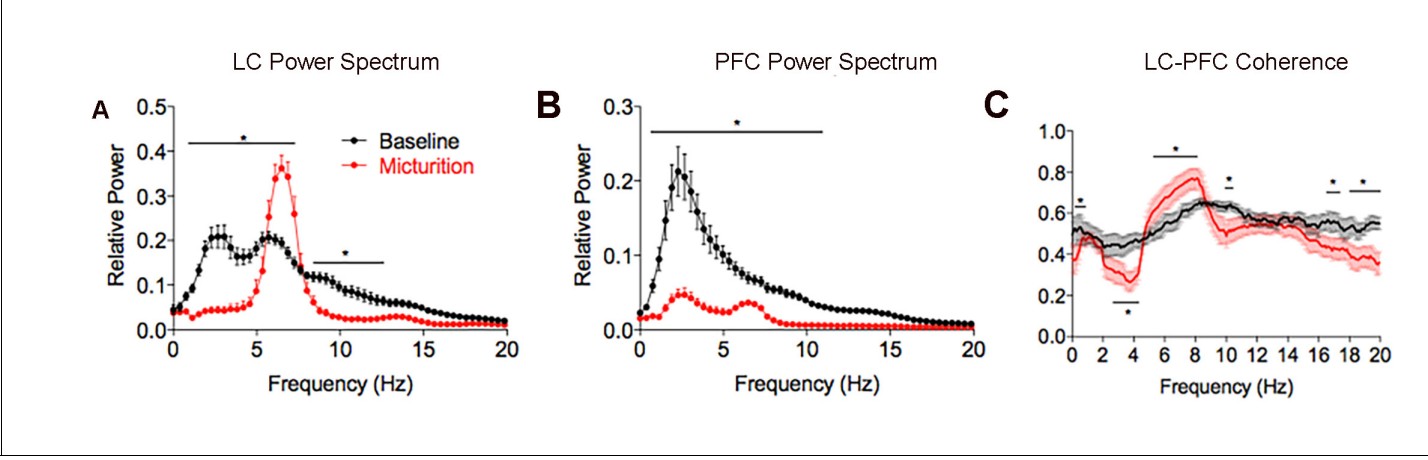

**Figure 6.** Mean power spectral density plots of LC and mPFC local field potentials and LC-mPFC coherence at different times preceding micturition. Red lines indicate the mean power (A,B) or coherence (C) determined 30 s before peak micturition pressure (red) and the black lines are the same for the preceding 30 s period. Means are from eight micturition cycles from three rats as described in the text. A two-way repeated measures ANOVA with frequency as the repeated measure indicated significant differences between the two time periods for LC power (F(1,742)=221, p<0.0001), PFC power (F(1,742)=671, p<0.0001 and LC-PFC coherence (F(1,2940) = 121, p<0.001).
DOI: https://doi.org/10.7554/eLife.29917.011

The present study is unique in recording single unit activity from the PMC in the intact unanesthetized and freely moving rat during cystometry. PMC neurons recorded in this study were homogeneous in the temporal correlation of their discharge pattern with the micturition cycle and resembled the previously described excitatory neurons. Notably, the present results are consistent with recent studies using photometry to measure the calcium currents in genetically-identified CRF-PMC neurons in unanesthetized mice during cystometry (*Hou et al., 2016*). These studies demonstrated a signal that was most prominent coincident with and up to 10 s after micturition. The present unit recordings have temporal resolution to detect changes in discharge rates and patterns such as bursts, that underlie the encoding of information. The post-micturition burst was consistent across all PMC neurons. If this results in increased glutamate release, it could be a mechanism to assure complete emptying. However, CRF is co-localized with glutamate in PMC neurons (*Hou et al., 2016*) and evidence suggests that CRF is inhibitory in this pathway and may counteract the excitatory effects of glutamate neurotransmission (*Pavcovich and Valentino, 1995*). Higher frequency bursts that favor neuropeptide release could be a mechanism for switching the function of the nucleus from one that facilitates active emptying to one that facilitates storage by countering detrusor contraction. Repetition of this high frequency burst during the intermicturition phases when bladder pressure is low would also be consistent with that. Future studies of simultaneous manipulation and recording of PMC neurons during cystometry would better address this question.

## LC neuronal activity

Rat LC neurons are relatively homogeneous in that they are all noradrenergic and have similar electrophysiological properties (*Aston-Jones et al., 1995*). A characteristic property of LC neurons of both anesthetized and unanesthetized rats is their activation by salient sensory stimuli of all modalities (*Aston-Jones and Bloom, 1981b*). LC neurons are also activated by diverse physiological stimuli including hypotension and non-noxious distention of the bladder or colon (*Elam et al., 1985*; *Elam et al., 1986*; *Valentino and Wehby, 1988*; *Page et al., 1992*; *Lechner et al., 1997*). In the present study, LC neuronal recordings during in vivo cystometry in unanesthetized rats revealed an exquisite sensitivity of LC neurons in that they become tonically activated approximately 20 s prior to micturition and in most cases before the steady pressure rise to micturition threshold (*Figures 3* and *5*). Possibly, this is a response to bladder wall stretching rather than an increase in intravesicular pressure. The timing of activation of LC neurons and PMC neurons with respect to micturition argues against our hypothesis that PMC projections to the LC drive its activation by bladder stimuli. This was particularly apparent in the one case in which both LC and PMC neurons could be recorded in

the same subject. However, because only a small subpopulation of PMC neurons project to the LC, it is possible that our neuronal sampling did not detect PMC-LC projection neurons (*Valentino et al., 1996*). Previous lesion studies provided evidence that LC activation by colonic distention was mediated by PMC but bladder distention was not examined in that study (*Rouzade-Dominguez et al., 2001*). Another route by which the LC could receive bladder information is through the ventrolateral periaqueductal gray (vlPAG). Anterogradely labeled vlPAG fibers innervate both PMC and the rostromedial peri-LC and synaptically contact LC dendrites in this region (*Bajic et al., 2000*). The PAG then could regulate both the LC and the PMC in response to bladder related stimuli. The apparent lack of temporal coordination between LC and PMC neuronal discharge could then be explained if they received distinct vlPAG afferents.

### Coordination between the bladder, LC and cortical activity

The LC-norepinephrine system densely innervates the mPFC and modulates its activity and executive functions (*Arnsten and Li, 2005*; *Arnsten, 2011*). Regionally selective pharmacological activation of the LC or activation using Designer Receptor Exclusively Activated by Designer Drug (DREADD) technology produces a desynchronization of the cortical EEG (an arousal response) (*Berridge and Foote, 1991*; *Vazey and Aston-Jones, 2014*). Moreover, LC activation by certain stimuli is necessary for their ability to elicit cortical arousal (*Valentino et al., 1991*). This includes bladder distention produced by injection of saline in anesthetized rats (*Page et al., 1992*). In addition to increasing arousal, the LC-norepinephrine system has been implicated in shifting attention towards salient stimuli (*Aston-Jones and Bloom, 1981b*; *Aston-Jones and Cohen, 2005*). The present results identified an important temporal relationship between LC activity, mPFC activity and bladder pressure, with LC and cortical activation being synchronized to occur approximately 20 s prior to micturition. The examination of network activity in the LC and between the LC and mPFC revealed that during this pre-micturition interval when LC neuronal activity is elevated, network activity shifts from low frequency oscillations to a prominent theta oscillation. At the same time mPFC activity desynchronizes as characterized by a decreased power in all frequencies but LC-mPFC coherence, specifically in the theta frequency, is increased. This network shift that is temporally offset from micturition may be a neural code to increase arousal and reset behavior prior to micturition so that voiding occurs in appropriate environments.

Taken together, the present results highlight the complex synchronization of pontine and cortical neurons with bladder afferent information that is required to maintain coordination between voiding behavior and urination. Just as normal bladder signals are relayed to the cortex, signals arising from bladder-related pathology could impact on cortical function through these same networks. Consistent with this, partial bladder outlet obstruction produces an abnormal urodynamic pattern that is associated with robust changes in cortical network activity (*Rickenbacher et al., 2008*). This condition is characterized by frequent non-micturition contractions that are temporally correlated with persistent cortical theta oscillations. Selective lesion of LC-forebrain projections using the toxin, DSP-4, prevented the cortical changes without affecting the abnormal urodynamics. This underscores the importance of the LC-cortical network in communicating bladder status to the cortex and suggests that dysregulation of this network could be a prominent signature of underlying bladder pathology.

## Materials and methods

### Subjects

Female adult Sprague-Dawley rats (Charles River, Wilmington, MA) were maintained in a temperature and light controlled environment (20°C, 12 hr light-dark cycle, lights on at 0700 hr) with food and water available *ad libitum*. Rats were housed 2/cage until the day of surgery. The care and use of animals were approved by the Children's Hospital of Philadelphia Institutional Animal Care and Use Committee.

### Surgery for electrode implantation

Surgery was performed at least 1 week after arrival at the animal facility. Surgery for implantation of an 8-microwire bundle electrode (NB Labs, Denison, TX) into the locus coeruleus (LC) was identical

to that previously described (*Curtis et al., 2012*). Briefly, rats were anesthetized with an isoflurane-air mixture and positioned in a stereotaxic frame. Body temperature was maintained at 37.5° C by a feedback controlled heating unit. A hole (4 mm diameter) was drilled in the skull centered at 3.7 mm caudal and 1.2 mm lateral to lambda for approaching the LC. Additional holes were drilled to insert skull screws for fixing the microwire bundle electrode to the skull with dental cement.

Neuronal recordings with glass micropipettes (2–4 µm diameter tip, 4–7 MOhm) filled with 0.5 M sodium acetate buffer were used to initially localize the LC. These were advanced toward the LC with a micromanipulator. Neuronal signals were amplified, filtered and monitored with an oscilloscope and a loudspeaker. LC neurons were tentatively identified during recording by their spontaneous discharge rates (0.5–5 Hz), entirely positive, notched waveforms (2–3 ms duration), and biphasic excitatory–inhibitory responses to contralateral hindpaw or tail pinch. Trajectories where LC units were encountered with the glass micropipette for at least 400 µm (dorsal-ventral penetration) were targeted for implantation with the microwire electrode bundle (NB Labs, Denison, TX) consisting of 8 Teflon insulated stainless steel wires (50 µm diameter) that were gathered in a circular bundle (7–8 mm long) and cut to produce bare wire tips for recording. A ground wire from the electrode encircled a skull screw and was in contact with brain tissue through another hole drilled next to the anchoring skull screw. The multiwire bundle was attached to a Microstar head stage and connected to a 16-channel data acquisition system (AlphaLab; Alpha Omega; Nazareth Illit, Israel). Accurate placement was aided by recording neuronal activity through the multiwire bundle during the implantation procedure. After detecting LC activity, the multiwire bundle was affixed to the skull and screws with dental cement. The scalp wound was sutured closed.

Surgery was similar for targeting of the electrode into the PMC, which is just rostral and medial to the ventral LC. As for the LC, the PMC was first localized with a glass micropipette as previously described. First the LC was localized and then the micropipette was repositioned rostrally until LC activity was no longer encountered. The multiwire bundle was then repositioned 200 µm medial to this point.

Rats had an additional depth electrode (tungsten microelectrode, 250 µm diameter) implanted into the medial prefrontal cortex (mPFC) (+3.2 AP, −0.6 ML, −3.0 DV) for mPFC local field potential (LFP) recordings. Animals were allowed 3 days to recover before bladder catheter implantation.

## Surgery for bladder catheter implantation

At least 2 days after the electrode implantation, rats were anesthetized with isoflurane and a small cut was made between the scapulae to provide an exit for the bladder catheter. A midline incision was made in the abdomen to access the bladder. A 5-French umbilical artery catheter was tunneled subcutaneously from the hole between the scapulae to the abdomen. The catheter end, which had been previously cauterized and flared, was brought intraperitoneally and inserted into the bladder dome and sutured above the flare. The exposed catheter end was connected to a port that allowed infusion of saline into the bladder.

## Experimental protocol

Three days after the bladder catheter implantation rats were placed into a cystometry chamber (Med Associates, St. Albans, CT). Sterile saline was continuously infused (100 µl/min) through the bladder catheter while urodynamic endpoints, including bladder pressure and the timing of micturition were recorded on-line for 1 hr using cystometry equipment (Medical Associates, St. Albans, VT) and software (Cystometry Analysis Software, SOF-522, Catamount R and D, St. Albans, VT). Neuronal activity was recorded from PMC neurons and/or LC neurons and LFPs were recorded simultaneously through micturition cycles. *Figure 1A* illustrates points along the pontine-cortical micturition circuit at which recordings were taken. Single unit LC waveforms were discriminated and sorted using the WaveMark template-matching algorithm in Spike2 (Cambridge Electronic Design, CED, v7.09, Cambridge, England), as described previously (*Curtis et al., 2012*). Local field potentials (LFPs) from LC were obtained from one of the wires of the multiwire bundle (sampled at 780 Hz, filtered from 1 to 150 Hz). Electrode recordings in the mPFC were amplified at a gain of 5000 and band pass filtered between 1–150 Hz.

## Neuronal firing analysis

LC neuronal firing rate (in Hz) was estimated in 5 s time bins as the total number of spikes in each time bin divided by duration of time bin in seconds. For each neuron, the z-score was calculated by subtracting the mean average background firing rate from the firing rate and then divided by the standard deviation. The examples showing the spike rasters are accompanied by histograms showing the z-score firing rate in 1 s bins. The average across animals were estimated in 5 s bins. For burst analysis the following criteria were used: the maximum interspike interval defining burst onset was 80 ms, interspike intervals $\leq$ 160 ms, number of spikes $\geq$ 4.

## Spectral analysis

To obtain time-frequency decomposition of the LFP signals, power spectral density was estimated using multitaper spectral estimators (Chronux toolbox scripts with MATLAB; three tapers, moving windows of 2 s width and 0.5 s overlap, $\pm1$ Hz bandwidth and displayed using interpolated smoothing between 0–20 Hz).

## Histology

At the end of the experiment, rats were anesthetized and current was passed through the electrode (10 mA, 15 s). Rats were transcardially perfused with 60 ml of 10% potassium ferrocyanide in 0.1 M phosphate buffered saline to form a Prussian blue reaction product for identification of the recording site. Frozen sections were cut on a cryostat and stained with neutral red for visualization of the Prussian blue labeled recording site (*Figure 1B*). Data were used only from histologically identified cases. There was no removal of outliers.

## Acknowledgements

This research was supported by DK102367, AG052780 and a Foederer grant.

## Additional information

### Funding

| Funder | Grant reference number | Author |
| --- | --- | --- |
| National Institute of Diabetes and Digestive and Kidney Diseases | DK102367 | Stephen A Zderic<br>Rita J Valentino |
| Foederer Foundation | | Stephen A Zderic<br>Rita J Valentino |
| National Institute on Aging | AG052780 | Stephen A Zderic<br>Rita J Valentino |

The funders had no role in study design, data collection and interpretation, or the decision to submit the work for publication.

### Author contributions

Anitha Manohar, Data curation, Formal analysis, Investigation, Methodology, Writing—original draft, Writing—review and editing; Andre L Curtis, Formal analysis, Supervision, Methodology, Writing—original draft, Supervision and training in LC recordings; Stephen A Zderic, Conceptualization, Methodology; Rita J Valentino, Conceptualization, Resources, Supervision, Funding acquisition, Writing—original draft, Project administration, Writing—review and editing

### Author ORCIDs

Anitha Manohar (iD) http://orcid.org/0000-0002-3011-2623
Rita J Valentino (iD) http://orcid.org/0000-0001-6839-6628

## Ethics

Animal experimentation: This study was performed in strict accordance with the recommendations in the Guide for the Care and Use of Laboratory Animals of the National Institutes of Health. All of the animals were handled according to approved institutional animal care and use committee (IACUC- protocol IAC12-000684) of the Children's Hospital of Philadelphia. All surgery was performed under isofluorane anesthesia. Animals received analgesics pre- and post-operatively and every effort was made to minimize suffering.

## Decision letter and Author response

Decision letter https://doi.org/10.7554/eLife.29917.014
Author response https://doi.org/10.7554/eLife.29917.015

## Additional files

### Supplementary files

• Transparent reporting form
DOI: https://doi.org/10.7554/eLife.29917.012

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
