## [Decision Letter]

Thank you for submitting your article "Network Dynamics Underlying The Encoding Of Bladder Information" for consideration by *eLife*. Your article has been reviewed by four peer reviewers, one of whom, Peggy Mason is a member of our Board of Reviewing Editors and the evaluation has been overseen by Sabine Kastner as the Senior Editor. The following individuals involved in review of your submission have agreed to reveal their identity: John T Williams (Reviewer #1); Joel Geerling (Reviewer #4).

The reviewers have discussed the reviews with one another and the Reviewing Editor has drafted this decision to help you prepare a revised submission.

This study provides novel and exciting insight into the hindbrain circuit activity supporting micturition in the behaving rat. The data are valuable. Please attend to the following major concerns and send us back a revision:

The figures need to be redone. They are difficult to view and do not do justice to the work.

Please give evidence that the LC and BAR neurons are indeed from LC and BAR. For LC, that is straightforward as the senior author is well aware. For BAR, provide all the evidence supporting the conclusion that neurons are from Bar and directly compare to the characteristics of BAR neurons in mice inferred in the Sabatini work.

There are some writing issues to fix which are detailed in the specific comments.

Reviewer #1:

This manuscript examines a sequence of events involving three brain areas correlating that activity in these areas with different phases of the micturition reflex. The results show that the activity in the locus coeruleus precedes an increase in firing in Barrington's nucleus and the activity in the locus coeruleus drives asynchronous activity in the prefrontal cortex. Upon micturition the activity in the locus coeruleus declines dramatically. The firing of neurons in Barrington's nucleus drives micturition. In addition, there are bursts of activity in Barrington's nucleus following micturition. The work defines a sequence of events resulting in a key biological process that requires a focused attention and a satisfying outcome.

1) Using awake behaving animals is a key step in obtaining the results.

2) The presentation of the results is stepwise and clear.

3) The numbers of animals is small but the clarity of the results do not justify or require a large number of animals. In this day of the requirement for sex differences in animal work some comment might be helpful. I can only imagine that it might be technically easier in females?

4) The central control of a very basic behavior such as micturition is not only important to understand but is also seemingly a great model. Although there may well be more components involved in this process this work defines key components.

Reviewer #2:

This is a very important and impressive investigation of the coordination between PFC and the pons in producing the full behavior of micturition. The authors recorded from PMC, LC (and in one case from both simultaneously) while recording LFPs in PFC and performing cystometry with infusion and a continuous record of vesicular pressure.

The result is the finding that PMC neurons are typically excited during emptying with a small excitation at the end of the void as well as not rare bursts during storage between voids. The LC neurons are excited in the moments leading up to emptying and so their change in activity precedes that of PMC cells. Results are consistent with the idea that LC activity desynchronizes / arouses PFC.

The authors point out both the limitations (e.g. recording bias away from some critical subpopulation of PMC cells) and the likely implications (e.g. PAG drives the show, first exciting LC cells and then PMC cells). The authors also make some intriguing speculations such as the idea that the bursts are releasing CRF important in storage whereas the emptying firing is such that it will result in Glu release and then excitation of sacral circuits.

My one major request is that the figures be improved. They are so small that even at >>100%, this reviewer cannot read them. Additionally, while the rasters are of use for many purposes, they do not allow the reader to see the firing rate (solid is solid). It is recommended that the authors put in instantaneous firing rate traces in place of or in addition to the rasters. Black on white is also recommended, again for better readability.

Reviewer #3:

The manuscripts is an original research article based on the recordings data from multiple sites within a pontine-cortical micturition circuit which included Barrington's nucleus (BN), the locus coeruleus (LC) and the medial prefrontal cortex (mPFC). The recordings were done simultaneously with in vivo cystometry performed to evaluate brain activity during micturition episodes in unanesthetized rats. The study is well designed and methodologically sound. It contains new information about temporal activation of different brain areas associated with distinct phases of the micturition cycle. Specifically, the authors established that there is a temporal offset between the shift in LC-mPFC network activity and micturition which allows to switch attention from ongoing activities to voiding behavior in a timely manner. A few points require additional clarifications.

1) Abstract. Please provide more specific information in the sentence "BN neurons had homogeneous firing patterns, characterized by tonic activity and phasic bursts…". What firing patterns were observed? What frequency? Amplitude etc.?

2)Results section. While it is understood that all the numerical parameters in this section are presented in Table 1, some numbers (# of bursts, spikes, duration etc.) and levels of statistical significance should be included in the text itself. When referring to "increased discharge rate…" please specify by how much, what percentage of neurons expressed this effect etc.

3) Figure 1 C2 panel – no numbers are visible on Y axis for MV and BP

4) Subsection “LC-cortical synchrony during micturition cycles*”* sentence "Interestingly, although the magnitude was small there was…" – small in comparison to what? And how small?

5) Discussion section – please remove "in the unanesthetized rat" as you mention it at the end of this sentence.

6) Discussion section describe the caveats of the study. If the listed caveats are applicable to the entire study (it looks like they are), then please move this paragraph to the end of the Discussion section.

7. Please proofread the text for misspellings etc. (there are at least 10 of them in the text).

Reviewer #4:

The authors provide interesting and potentially useful new information in rats correlating the timing of micturition and neuronal firing in a brainstem region as well as LFP in the mesial frontal cortex. A strength of their work is the combination of neuronal recordings and bladder cystometry in awake animals, and their attempting to record multiple groups of neurons. A weakness of the paper is the lack of any experiment attempting to test their hypothesis that these neurons are in fact connected (that activity in one can be triggered or interrupted by manipulating the others) or that any of this activity influences behavior in the way they hypothesize at the end of the abstract. There are some uncertainties inherent to their approach, detailed below, which undercut the certainty expressed in conclusions throughout the text (see below).

More importantly, after comparing their results with previous examples of micturition-related unit activity in the dorsolateral pons (including results published from the same lab) I am not convinced that they recorded neurons inside the specific, tiny populations they claim to have stabbed with their electrodes. Finding LC-like activity and then targeting that region (or something ventral-medial to that region) as described in the methods did get their electrodes into the right ballpark, but from there, based on the histology and unit data shown in the figures, it does not appear that neurons in what they call Barrington's nucleus, or having any of the characteristics of a BN neuron (firing just before and during micturition, causing bladder contraction) were recorded in the "BN" unit group. Instead, some or most of the units labeled "LC" appear likely to represent neuronal activity of BN neurons. Showing electrode tracts from exemplary cases in neutral-red (Nissl) stained tissue is better than not showing anything, but it does not reveal the neurons that were recorded or in any way support the claim that any of them were part of the now-well-defined populations that the authors were attempting to record. Previous work from the authors used juxtacellular labeling (Rouzade-Dominguez et al., 2003) to reveal the location and peptidergic identity of neurons in this region, and newer molecular-genetic techniques allow definitive identification using Cre-conditional optrode stimulation/recording or calcium imaging. Lacking any of these more specific methods for cellular identification, suppositional labels like "LC" or "BN" are unwarranted -- these labels should be dropped, especially given that the only units with a BN/PMC-like pattern of activity (beginning prior to bladder contraction, increasing as bladder pressure rises, and then stopping ~once the sphincter opens and bladder pressure falls) were grouped as "LC".

Another concern I had relates to the numbers of animals and numbers of units in each analysis. The animal N and unit n in certain analyses from the Results section through the Materials and methods was left a bit hazy, and should be clarified.

---

## [Author Response]

This study provides novel and exciting insight into the hindbrain circuit activity supporting micturition in the behaving rat. The data are valuable. Please attend to the following major concerns and send us back a revision:The figures need to be redone. They are difficult to view and do not do justice to the work.Please give evidence that the LC and BAR neurons are indeed from LC and BAR. For LC, that is straightforward as the senior author is well aware. For BAR, provide all the evidence supporting the conclusion that neurons are from Bar and directly compare to the characteristics of BAR neurons in mice inferred in the Sabatini work.

Figures. The figures have been redone as suggested. Figure 1, which had several parts was divided into two figures and a supplemental figure. The lettering on the axes have been made larger. The traces are now black on white and records of spontaneous rate have been added. A reason for not including the ratemeter records originally is that PMC neurons dramatically increase rates during bursts and rate plots don’t have the resolution to show the details of bursts or even that the cells show occasional tonic activity because it is relatively so low, i.e., the pattern and low firing rate activity is lost in the ratemeter record when there are short duration high frequency bursts. Therefore, we kept rasters and raw waveform traces. The raster in Figure 2 is now black on white. The original Figure 4 was divided into two figures. The lettering on the axes was enlarged. We want to emphasize that our goal in most cases was to show as much data from individual neurons as possible rather than means because this underscores how firing rates and patterns were consistently affected by changes in bladder pressure in a particular nucleus.

Electrode localization. The terms Barrington’s nucleus (BN) and the pontine micturition center (PMC) have been used interchangeably to refer to the area in the dorsolateral pons containing neurons that project to the lumbosacral spinal cord which when electrically or chemically stimulated elicit bladder contraction. However, neurons in this region that are transsynaptically labeled from the bladder extend outside of the counterstained oval cluster that has come to be known as the BN. Given this and the comments of Reviewer 4 we replaced our reference to BN with the PMC and this is also consistent with the nomenclature used by Hou et al., (2016). Chronic recordings in unanesthetized rats require multiwire microelectrodes. A limitation with recordings from multiwire electrodes is that one cannot precisely identify the neurons being recorded. Unlike experiments in anesthetized head-fixed rats in which neurons can be recorded with glass microelectrodes filled with biocytin, one cannot juxtacellularly label neurons with wire electrodes and determine their identity later. And the limitation to recordings in anesthetized rats is that anesthesia confounds the micturition reflex so this experiment could not have been done using juxtacellular labeling. It is not valid to compare the characteristics of the CRF+PMC neurons recorded in the paper by Hou et al., (2016) to our PMC neurons for two reasons. Neural recordings made from electrodes in the Hou et al., (2016) paper were whole cell recordings done in slices and activity was evoked by current injection. The in vivo recordings performed during micturition in the Hou paper used fiber photometry which measures calcium transients, not action potentials. This type of recording does not detect rate or pattern and cannot detect bursts that we see in our electrode recordings. Nonetheless, their findings are notably consistent with our data in showing that the largest activation of PMC neurons occurs following (and not before) the peak bladder pressure for several seconds and this similarity is now highlighted the Discussion section of the revised manuscript.

We have recorded from locus coeruleus (LC) neurons in anesthetized and unanesthetized rats for over 30 years and are familiar with the distinct electrophysiological characteristics of the neurons which we identify prior to securing the electrode. These are described in the Matierials and methods. During recordings in unanesthetized rats, LC neurons show a characteristic positive correlation with the arousal state as indicated by cortical electroencephalographic or local field potential (LFP) activity (Aston-Jones and Bloom, 1981). This was an important aspect of our findings, where bladder pressure served as the eliciting stimulus. However, it can also be seen in our traces at other times, i.e., when LC discharge rate is low, cortical LFP activity is synchronous and of high amplitude and low frequency, consistent with low arousal state. When LC discharge is high, cortical activity becomes desynchronized (low amplitude, high frequency) characteristic of arousal. This is very characteristic of LC neurons and was observed in all of our LC traces. These electrophysiological characteristics coupled with the histology provide evidence that our LC recordings were indeed in LC.

There are some writing issues to fix which are detailed in the specific comments.Reviewer #1

We appreciate the reviewer’s comments. The only comment requiring an answer was #3.

3) The numbers of animals is small but the clarity of the results do not justify or require a large number of animals. In this day of the requirement for sex differences in animal work some comment might be helpful. I can only imagine that it might be technically easier in females?

The reviewer is correct in that the bladder surgery is technically easier for female rats and that was a rationale for the use of females only. However, in past studies in which we performed cystometry and cortical LFP recordings in unanesthetized male rats, we also demonstrated that cortical desynchronization preceded peak micturition pressure and micturition (Kiddoo et al., 2006; Rickenbacher et al.,2008). We discuss the use of a single sex in the Caveat section.

Reviewer #2:This is a very important and impressive investigation of the coordination between PFC and the pons in producing the full behavior of micturition. The authors recorded from PMC, LC (and in one case from both simultaneously) while recording LFPs in PFC and performing cystometry with infusion and a continuous record of vesicular pressure.The result is the finding that PMC neurons are typically excited during emptying with a small excitation at the end of the void as well as not rare bursts during storage between voids. The LC neurons are excited in the moments leading up to emptying and so their change in activity precedes that of PMC cells. Results are consistent with the idea that LC activity desynchronizes / arouses PFC.The authors point out both the limitations (e.g. recording bias away from some critical subpopulation of PMC cells) and the likely implications (e.g. PAG drives the show, first exciting LC cells and then PMC cells). The authors also make some intriguing speculations such as the idea that the bursts are releasing CRF important in storage whereas the emptying firing is such that it will result in Glu release and then excitation of sacral circuits.My one major request is that the figures be improved. They are so small that even at >>100%, this reviewer cannot read them. Additionally, while the rasters are of use for many purposes, they do not allow the reader to see the firing rate (solid is solid). It is recommended that the authors put in instantaneous firing rate traces in place of or in addition to the rasters. Black on white is also recommended, again for better readability.

We appreciate the reviewer’s comments. The major criticism related to the figures and these were revised as described above.

Reviewer #3:The manuscripts is an original research article based on the recordings data from multiple sites within a pontine-cortical micturition circuit which included Barrington's nucleus (BN), the locus coeruleus (LC) and the medial prefrontal cortex (mPFC). The recordings were done simultaneously with in vivo cystometry performed to evaluate brain activity during micturition episodes in unanesthetized rats. The study is well designed and methodologically sound. It contains new information about temporal activation of different brain areas associated with distinct phases of the micturition cycle. Specifically, the authors established that there is a temporal offset between the shift in LC-mPFC network activity and micturition which allows to switch attention from ongoing activities to voiding behavior in a timely manner. A few points require additional clarifications.1) Abstract. Please provide more specific information in the sentence "BN neurons had homogeneous firing patterns, characterized by tonic activity and phasic bursts…". What firing patterns were observed? What frequency? Amplitude etc.?.

The limitation to 150 words in the Abstract really precludes details. It is also important to note that because these are extracellular recordings amplitude is not meaningful because it is related to the proximity of the cell to the electrode, it is not a true action potential amplitude which could only be determined from intracellular recordings. Also firing rates for PMC neurons are not very meaningful because the neurons don’t have a tonic regular firing rate.

2)Results section. While it is understood that all the numerical parameters in this section are presented in Table 1, some numbers (# of bursts, spikes, duration etc.) and levels of statistical significance should be included in the text itself. When referring to "increased discharge rate…" please specify by how much, what percentage of neurons expressed this effect etc.

We now indicate that all PMC neurons showed burst activity and added the mean frequency and duration in the text. Because basal discharge rates are variable we plotted the rates for each individual neuron that we recorded from and this is shown in Figure 2 with an overall statistic for each animal so the reviewer and readers can see what happened to every cell. This is much more revealing than documenting an average rate in the text.

3) Figure 1 C2 panel – no numbers are visible on Y axis for MV and BP

This has now been added and the Figure is now Figure 1—figure supplement 1.

4) Subsection “LC-cortical synchrony during micturition cycles” sentence "Interestingly, although the magnitude was small there was…" – small in comparison to what? And how small?

This was addressed in comment of reviewer 2 above (Discussion section).

5) Discussion section – please remove "in the unanesthetized rat" as you mention it at the end of this sentence.

Corrected.

6) Discussion section describe the caveats of the study. If the listed caveats are applicable to the entire study (it looks like they are), then please move this paragraph to the end of the Discussion section.

This is a matter of style and we prefer to begin a discussion of the interpretation of our results by first stating the caveats.

7. Please proofread the text for misspellings etc. (there are at least 10 of them in the text)

Done.

Reviewer #4:The authors provide interesting and potentially useful new information in rats correlating the timing of micturition and neuronal firing in a brainstem region as well as LFP in the mesial frontal cortex. A strength of their work is the combination of neuronal recordings and bladder cystometry in awake animals, and their attempting to record multiple groups of neurons. A weakness of the paper is the lack of any experiment attempting to test their hypothesis that these neurons are in fact connected (that activity in one can be triggered or interrupted by manipulating the others) or that any of this activity influences behavior in the way they hypothesize at the end of the abstract. There are some uncertainties inherent to their approach, detailed below, which undercut the certainty expressed in conclusions throughout the text (see below).

This is the first study to combine neuronal recordings in these multiple brain regions simultaneously with measurements of urodynamic function in unanesthetized freely moving animals and that is its strength. We discuss how the data (with caveats) argue against the hypothesis that Barrington’s nucleus is signaling bladder information to the LC and this was important. As an alternative we speculate that LC activation prior to micturition may be through periaqueductal gray afferents. However, the examination of other afferents to the LC was beyond the scope of this manuscript and the subject of future studies. There is a well-established literature that the LC projects to the mPFC and modulates its activity. With regards to bladder related activity, we previously demonstrated that LC activation was required for cortical activation evoked by bladder distention in anesthetized rats (Page et al., 1992) and that is now mentioned in the Discussion section. We also discuss our previous study that provided evidence that the LC was necessary for the development of cortical theta oscillations in rats with partial bladder outlet obstruction (Rickenbacher et al., 2008). Along with this literature, our demonstration in the present study of the consistent temporal correlation between LC and mPFC activation during the micturition cycle, accompanied by increased LC-mPFC coherence at the time preceding micturition support the hypothesis that LC activation initiates arousal prior to micturition.

More importantly, after comparing their results with previous examples of micturition-related unit activity in the dorsolateral pons (including results published from the same lab) I am not convinced that they recorded neurons inside the specific, tiny populations they claim to have stabbed with their electrodes. Finding LC-like activity and then targeting that region (or something ventral-medial to that region) as described in the methods did get their electrodes into the right ballpark, but from there, based on the histology and unit data shown in the figures, it does not appear that neurons in what they call Barrington's nucleus, or having any of the characteristics of a BN neuron (firing just before and during micturition, causing bladder contraction) were recorded in the "BN" unit group. Instead, some or most of the units labeled "LC" appear likely to represent neuronal activity of BN neurons. Showing electrode tracts from exemplary cases in neutral-red (Nissl) stained tissue is better than not showing anything, but it does not reveal the neurons that were recorded or in any way support the claim that any of them were part of the now-well-defined populations that the authors were attempting to record. Previous work from the authors used juxtacellular labeling (Rouzade-Dominguez et al., 2003) to reveal the location and peptidergic identity of neurons in this region, and newer molecular-genetic techniques allow definitive identification using Cre-conditional optrode stimulation/recording or calcium imaging. Lacking any of these more specific methods for cellular identification, suppositional labels like "LC" or "BN" are unwarranted -- these labels should be dropped, especially given that the only units with a BN/PMC-like pattern of activity (beginning prior to bladder contraction, increasing as bladder pressure rises, and then stopping ~once the sphincter opens and bladder pressure falls) were grouped as "LC".

I respectfully argue with the Reviewer’s suggestion that the neurons we indicated as LC neurons were actually Barrington’s nucleus neurons. To reiterate, our LC recordings were in the LC based not only on histological verification, but on the neuronal discharge characteristics in unanesthetized rats described by our laboratory and others (Aston-Jones and Bloom, 1981). Importantly, BN neurons would not be expected to exhibit a regular tonic discharge rate in the absence of bladder pressure increases followed by an activation 20-30 s prior to micturition as was consistently observed in the LC neurons of this study. Conversely LC neurons do not go from being nearly silent to a 25 Hz burst of 600 ms duration like the neurons we indicate are BN neurons. This is not at all characteristic of LC neurons. Even when the bladder is mechanically distended to bring pressure up to 70 mm Hg (greater than in this study), LC neuronal discharge only increases to 50-100% of its baseline rate (Elam et al., 1986; Page et al., 1992). Finally, as reproduced above, the findings of Hou et al., 2017 using optical recordings from genetically-identified CRF+ BN neurons were consistent with ours by showing activation with the onset and following micturition, not 20-30 s before.

Another concern I had relates to the numbers of animals and numbers of units in each analysis. The animal N and unit n in certain analyses from the Results through the Methods was left a bit hazy, and should be clarified.

We included the number of cells and rats in all analyses.